# Unlocking Overexpressed Membrane Proteins to Guide Breast Cancer Precision Medicine

**DOI:** 10.3390/cancers16071402

**Published:** 2024-04-03

**Authors:** Júlia Badaró Mendonça, Priscila Valverde Fernandes, Danielle C. Fernandes, Fabiana Resende Rodrigues, Mariana Caldas Waghabi, Tatiana Martins Tilli

**Affiliations:** 1Translational Oncology Platform, Center for Technological Development in Health, Fundação Oswaldo Cruz (Fiocruz), Rio de Janeiro 21040-900, RJ, Brazil; juliamendonca@aluno.fiocruz.br; 2Laboratory of Applied Genomics and Bioinnovation, Instituto Oswaldo Cruz (IOC) Fiocruz, Rio de Janeiro 21045-900, RJ, Brazil; mariana@ioc.fiocruz.br; 3Divisão de Patologia (DIPAT), Instituto Nacional de Câncer (INCA), Rio de Janeiro 20230-130, RJ, Brazil; pvalverde@inca.gov.br (P.V.F.); dfernandes@inca.gov.br (D.C.F.); fabiana.resende@inca.gov.br (F.R.R.); 4Laboratory of Clinical and Experimental Pathophysiology, IOC, Fiocruz, Rio de Janeiro 21041-210, RJ, Brazil

**Keywords:** breast cancer, therapy, diagnosis, molecular targets, precision oncology

## Abstract

**Simple Summary:**

Breast cancer (BC) is a global health concern, hindered by the limited effectiveness and adverse effects of current treatments. To address this, we propose a method for identifying membrane proteins in tumors, offering potential targets for BC therapy and diagnosis by analyzing gene expressions in breast tumor and healthy tissues using bioinformatics tools, like TCGA, UALCAN, TNM Plot, and LinkedOmics. Four transcripts (LRRC15, EFNA3, TSPAN13, and CA12) were identified with heightened expressions in BC tissue. These transcripts showed high accuracy in identifying tumor samples and were consistently elevated across all BC molecular subtypes. Tissue microarray (TMA) analysis confirmed an increased expression in tumor tissues compared to adjacent breast tissue. The study underscores the potential of LRRC15, EFNA3, TSPAN13, and CA12 as biomarkers for enhancing BC diagnosis and as promising therapeutic targets with reduced side effects and improved efficacy.

**Abstract:**

Breast cancer (BC) is a prevalent form of cancer affecting women worldwide. However, the effectiveness of current BC drugs is limited by issues such as systemic toxicity, drug resistance, and severe side effects. Consequently, there is an urgent need for new therapeutic targets and improved tumor tracking methods. This study aims to address these challenges by proposing a strategy for identifying membrane proteins in tumors that can be targeted for specific BC therapy and diagnosis. The strategy involves the analyses of gene expressions in breast tumor and non-tumor tissues and other healthy tissues by using comprehensive bioinformatics analysis from The Cancer Genome Atlas (TCGA), UALCAN, TNM Plot, and LinkedOmics. By employing this strategy, we identified four transcripts (LRRC15, EFNA3, TSPAN13, and CA12) that encoded membrane proteins with an increased expression in BC tissue compared to healthy tissue. These four transcripts also demonstrated high accuracy, specificity, and accuracy in identifying tumor samples, as confirmed by the ROC curve. Additionally, tissue microarray (TMA) analysis revealed increased expressions of the four proteins in tumor tissues across all molecular subtypes compared to the adjacent breast tissue. Moreover, the analysis of human interactome data demonstrated the important roles of these proteins in various cancer-related pathways. Taken together, these findings suggest that LRRC15, EFNA3, TSPAN13, and CA12 can serve as potential biomarkers for improving cancer diagnosis screening and as suitable targets for therapy with reduced side effects and enhanced efficacy.

## 1. Introduction

Breast cancer (BC) is the most common cancer among women globally and is the leading cause of cancer-related deaths in women, corresponding to 15.5% of annual cancer deaths [1]. BC exhibits varying levels of tumor aggressiveness, behaviors, therapy susceptibility, and clinical prognosis due to its heterogeneous histological, biological features and molecular subtypes [2]. These subtypes are grouped into four main categories based on both molecular and histological evidences: the BC-expressing hormone receptor (estrogen receptor (ER^+^) or progesterone receptor (PR^+^), which are classified as luminals A and B, respectively), BC-expressing human epidermal receptor 2, classified as HER2^+^ tumors, and triple-negative breast cancer (TNBC) (ER^−^, PR^−^, HER2^−^) [3,4]. The molecular classification is important to identify and categorize patients who may benefit from targeted therapy, such as hormone therapy and anti-HER2 therapy. TNBC constitutes 20% of all breast tumors and is characterized by its aggressiveness, early relapse, and is the major subtype observed in advanced stages. Based on the absence of hormone receptors and HER-2 expression, TNBC patients cannot benefit from target therapy; therefore, chemotherapy is the pillar of treatment. Despite advancements in precision clinical care, many BC patients face long-term treatment side effects, and some eventually experience a relapse or drug resistance [5,6]. Moreover, BC metastasis accounts for 90% of mortality and our understanding of metastasis remains limited [7,8].

The identification of the molecular mechanisms that lead to tumorigenesis and cancer progression represents a critical step for developing more effective therapies, improving diagnoses, and establishing correlations between clinical behavior and disease etiology. With the methodological advances achieved in recent decades, the implementation of the use of high-throughput omics technologies, such as genomics, proteomics, transcriptomics, metabolomics, and microbiomics, have revolutionized cancer research [9,10]. Genomics and transcriptomics, for instance, enable the identification of genetic alterations, elucidation of cancer genome structure, and discovery of differentially expressed genes involved in cancer progression and maintenance [11]. Additionally, the combination of mRNA profiling with established genomic platforms offers several benefits in clinical diagnosis and improving the accuracy in classifying BC subtypes. This integration simplifies the identification of specific groups that would respond favorably to personalized therapeutic interventions adapted to their individual characteristics, thus optimizing patient care and outcomes [12].

All these technological advances improve the discovery of biomarkers, mainly based on cancer-specific cellular alterations, of which membrane proteins are of great interest. These proteins play crucial roles in cell signaling processes, including cell–cell interactions, and cell environment sensing, collecting cell signals and transmitting them to the interior of the tumoral cell, and, moreover, membrane proteins have been recognized as major drug targets, due to their accessibility [13]. Currently, membrane proteins represent about 60% of drug targets on the market [14]. Identifying these membrane biomarkers can enhance the development of diagnostic tools for tumor tissues and circulating tumor cells, aiding in accurate therapeutic strategies, which are urgently required [15,16,17].

In the present study, we proposed an experimental strategy for identifying upregulated transcripts encoding membrane proteins by using comprehensive bioinformatics analyses in several databases. Thereby, we identified four overexpressed transcripts in BC samples: Leucine-rich 15 repeat protein (LRRC15), Ephrin A3 (EFNA3), Tetraspanin 13 (TSPAN13), and Carbonic Anhydrase XII (CA12). Furthermore, we validated the overexpression of these four biomarkers at the protein level and in BC clinical samples tissues. We believe that LRRC15, EFNA3, TSPAN13, and CA12 can improve BC early diagnosis and aid in the design of new personalized drugs with impacts on the minimization of side effects.

## 2. Materials and Methods

### 2.1. Data Processing, Samples, and Databases

The study was separated into two cohorts named as the discovery set and validation set. The mRNA expression profiles (RNAseqV2) and correlative clinical data from 1102 cases of BC samples and corresponding normal samples were downloaded from TCGA (http://cancergenome.nih.gov/ (accessed on 27 July 2018). The study flowchart is presented in Figure 1.

Discovery set:

The discovery set cohort included 111 paired RNA-seq data from BC tumor and non-tumor tissues, and the graphs were generated using TNMplot. Additionally, the transcriptome data were also obtained from other non-tumor tissues, comprising the bladder (*n* = 19), lung (*n* = 59), pancreas (*n* = 4), uterus (*n* = 13), colon (*n* = 40), kidney (*n* = 32), stomach (*n* = 32), esophagus (*n* = 11), liver (*n* = 50), and head and neck (*n* = 43).

Validation set:

The validation set cohort included 991 tumor samples. For the expression profile analysis of the different molecular subtypes of BC, the 933 patients (58 patients were excluded from the study due to a lack of information on the histopathological and molecular data) were separated based on the main molecular subtypes: luminal A (*n* = 475), luminal B (*n* = 227), triple negative (*n* = 163), and enriched HER2 (*n* = 68).

A second step of validation was conducted using the data from gene chip data accessed using the TNMplot tool (https://www.tnmplot.com/ (accessed on 14 January 2022). This web platform allows the analysis of differential gene expressions in 7569 malignant breast tumor tissues compared with 242 normal breast tissues and 82 metastasis samples [18].

Validation at the protein level was accessed from the UALCAN database (https://ualcan.path.uab.edu/analysis-prot.html (accessed on 10 August 2021)) [19]. UALCAN is an interactive web resource for analyzing protein expression analyses from the Clinical Proteomic Tumor Analysis Consortium (CPTAC), and clinicopathological data are included [20].

To explore the single-cell RNA (scRNA) gene expression of the four targets in PBMCs, we used the public HPA database (proteinatlas.org (accessed on 21 April 2022)) [21]. This analysis allowed us to evaluate the expression profiles of the four potential targets in other samples to characterize their expression specificity for breast tumors.

### 2.2. Identification of Overexpressed Transcripts Encoding Membrane Proteins

The transcripts that had differential expression profiles between the tumor sample and its paired non-tumor control tissue were identified using the following factors: (i) cut-off of a 2-fold expression level increase in tumor samples as compared to non-tumoral breast samples; (ii) selection of transcripts’ encoding membrane proteins, which were identified using the UniprotKb (http://www.uniprot.org/ (accessed on 23 November 2018)) and The Human Protein Atlas (http://www.proteinatlas.org/ (accessed on 23 November 2018)) databases; and (iii) low expression in non-tumoral tissues other than breast tissue.

### 2.3. Protein–Protein Interaction (PPI) Network

To investigate the possible intracellular signaling events in which the selected membrane proteins could be involved, we conducted a study of the interactions of the identified proteins obtained from the GeneMANIA database (http://www.genemania.org (accessed on 16 October 2023)). These databases provide concise PPIs utilizing an extensive repository of functional association data to identify genes closely linked to a given set of input genes. An interesting aspect of the GeneMANIA process involves the determination of network weights to establish the strength of these connections [22]. The studied genes were indicated with stripes. We used the 10-gene filter with physical associations and automatically selected the weighting method.

We employed LinkedOmics (http://www.linkedomics.org/ (accessed on 29 September 2023)) and LinkInterpreter modules to evaluate the pathway enrichment through the Gene Set Enrichment Analysis (GSEA).

### 2.4. Analysis of the Diagnostic Value of the Selected Targets

To determine the sensitivity and specificity of the selected transcripts to diagnosis breast tumor, receiver operating characteristic curves (ROCs) were produced. The area under the ROC curve (AUC), with 95% confidence intervals (CIs), was calculated for each protein using Prism 8.0 software (Graphpad Prism/ Boston, MA). The optimal cut-off thresholds were determined by using the highest Youden index, based on the point at which the sensitivity + specificity were the maximum. In all curves, *p* ≤ 0.05 was considered statistically significant.

### 2.5. Pan-Cancer Analysis of the Selected Genes in Tumor and Adjacent Non-Tumor Tissues

The Tumor Immune Estimation Resource 2.0 (TIMER2.0; http://timer.cistrome.org/ (accessed on 22 April 2022)) website is user-friendly and an interactive web resource for analyzing cancer TCGA data for tumors compared to adjacent non-tumor tissue. In our study, we chose the “Gene_DE” option to investigate differential expressions of LRRC15, EFNA3, TSPAN13, and CA12 in the tumors of 33 neoplastic tissues from TCGA.

### 2.6. Tissue Microarray

BC tissue microarray slides (US Biomax, ref.: BC081116d/ (Rockville, MD, USA)) were kept at 60 °C for 2 h before use. For the deparaffinization step, the slides were bathed in xylol (3× for 5 min); then, the sections were dehydrated in decreasing concentrations of ethanol (100, 95, 80, and 70%—5 min each). Then, the tissues were hydrated in water for 5 min. For antigenic recovery, the slides were placed in a streamer containing a Citrate pH 6.0/Tris EDTA pH 9.0/Trilogy™ (Cell Marque, La Marque, TX, USA) solution for 30 min. Then, after being washed with TBS (3× for 5 min), the nonspecific binding was blocked with the Novolink™ Protein Block (Leica Biosystems, Sao Paulo, Brazil) for 5 min. Subsequently, the slides were incubated overnight with the following primary antibodies: Anti-LRRC15 (1:400; Sigma-Aldrich, St. Louis, MO, USA), Anti-EFNA3 (1:400; Sigma-Aldrich), Anti-TSPAN13 (1:200), and Anti-CA12 (1:200; Abcam, Cambridge, UK) (Appendix A). After washing with distilled water, the slides were treated for 30 min with the post-primary and polymer reagents from the Novolink kit, followed by three washes with TBS. Positive reactions were visualized using the DAB (3,3’-diaminobenzidine) solution and counterstained with hematoxylin.

The TMA slides comprised a collection of 100 cases of Invasive breast carcinomas, 9 adjacent normal breast tissues, and 1 adjacent breast tumor tissue, all accompanied by relevant patient clinical data. The immunostained slides were evaluated and scored by a pathologist. The protein staining intensity for each target was subsequently categorized into four groups based on the percentage of tumor cells exhibiting positive staining:0: incomplete, weak, and scanty staining in the membranes of <10% of tumor cells.1+: incomplete, weak, and scanty staining in the membranes of >10% of tumor cells.2+: circumferential and incomplete staining and/or weak/moderate membrane staining in more than 10% of tumor cells, or complete and/or intense membrane staining in more than 10% of tumor cells.3+: uniform and intense membrane staining of tumor cells.

### 2.7. Statistical Analysis

Statistical analyses were performed and graphical data were obtained using Prism 8.0 software (Graphpad). Quantitative data were analyzed by the parametric Student’s *t*-test to compare them the different expression levels between breast tumor samples and paired to non-tumor samples, and non-parametric tests for other analyses are indicated in the figure legends. Differences were considered statistically significant when the *p*-values were * *p* < 0.05, ** *p* < 0.01, and *** *p* < 0.001.

## 3. Results

### 3.1. Clinical, Pathological, and Molecular Characteristics of the Cohort from TCGA Database

TCGA database cohort obtained for this study was divided into two groups: the discovery set (DS) and the validation set (VS). The DS consisted of 111 paired samples, including tumor and non-tumor samples from the same patient, with representations from various subtypes of BC. Specifically, within these 111 samples, the distribution of breast cancer subtypes was as follows: 55% luminal A, 20% luminal B, 13% HER2-positive, and 12% triple-negative breast cancer (TNBC). The VS included an expanded cohort with 991 BC samples. The clinical and pathological characteristics of both cohorts are summarized in Table 1. In summary, both cohorts presented an average age of 51 to 70 years (50%), predominantly postmenopausal women (68%), diagnosed with grade II tumors (57%), with TNM classifications, T2 (58%), N0 (47%), and M0 (83%), and classified as luminal molecular subtypes (75%).

### 3.2. LRRC15, EFNA3, TSPAN13, and CA12 Are Highly Expressed in BC Patients

To identify differentially expressed genes that encode membrane proteins between BC and normal breast samples, an RNA expression analysis was performed on 111 tumor samples representing multiple molecular subtypes and 111 adjacent non-tumoral breast human tissues from TCGA database. Out of the 20.523 genes included in the transcriptome files, 90 genes encoding membrane proteins were identified as upregulated in BC. The criteria for the transcript selection were as follows: (i) at least a 2-fold expression increase in tumor samples compared to non-tumoral breast samples, (ii) transcripts encoding membrane proteins, and (iii) a low expression in non-tumoral tissues other than breast tissue. Based on these factors, LRRC15, EFNA3, TSPAN13, and CA12 were identified as transcripts with higher expression and prevalence levels in breast tumors, with low expressions in most of the control tissues. Figure 2 shows the increased gene expression profiles of these four transcripts compared to non-tumor breast tissue. Among them, LRRC15 presented the higher expression levels, reaching an 18-fold increase compared to non-tumor breast tissue (Figure 2A), while EFNA3, TSPAN13, and CA12 showed approximately a 4-fold increase (Figure 2B–D). In an expanded analysis, the gene expressions of the four selected transcripts were significantly higher in breast tumor samples compared to non-tumor samples from other tissues (Figure 2E–H), with a few exceptions. For example, the EFNA3 transcript showed a high expression in non-tumoral head and neck tissues (Figure 2F), and CA12 was highly expressed in non-tumoral colon and kidney tissues (Figure 2H).

### 3.3. Validation of LRRC15, EFNA3, TSPAN13, and CA12 Overexpressions in BC Tissues

Based on the results obtained for the DS cohort, we next proposed to expand the gene expression profile to a VS cohort, comprising 991 patients with breast tumor. Overall, the increased expressions of LRRC15, EFNA3, TSPAN13, and CA12 were statistically significant between DS and VS as compared to the control samples (Figure 3A–D). In summary, these results validate the overexpression of the selected transcripts with specificity to breast tumors.

Next, we performed a second validation of the differential expressions of the four targets using the TNMplot database. This inquiry utilized a genetic chip-based approach, comparing gene expressions across malignant breast tumors (*n* = 7569), metastatic samples (*n* = 82), and non-tumoral samples (*n* = 242). Our analysis consistently confirmed the increased expressions of LRRC15, EFNA3 TSPAN13, and CA12 in tumor samples, consistent with TCGA data findings from the DS and VS cohorts. Moreover, metastatic samples exhibited elevated transcript levels compared to non-tumoral samples, except for CA12 (Figure 3E–H). Comparing the tumor samples with metastatic ones, we observed significant results for LRRC15, TSPAN13, and CA12 as determined by Dunn’s test (*p* < 0.05). This significant increased expression of the four targets at metastatic breast cancer sites suggests their potential as biomarkers for clinical progression or as therapeutic candidates for advanced disease stages.

### 3.4. Potential Use of the Four Targets for BC Molecular Classification and Precision Diagnosis

BC, specifically, exhibits significant heterogeneity from the onset of the disease and can be clinically classified into four main molecular profiles: luminal A, luminal B, basal-like (predominantly TNBC), and HER2-enriched. Among these profiles, TNBC is considered the most aggressive type of cancer with high rates of tumor relapse [23,24]. In this study, we investigated whether there were differences in the gene expression profiles of the target genes among the molecular subtypes. Our findings reveal that the transcript CA12 is highly expressed in the luminal A and luminal B groups compared to the control, TNBC, and HER2-enriched (Figure 4D). Conversely, the transcript TSPAN13 showed a higher expression in the luminal A, luminal B, and HER2-enriched groups (Figure 4C). Both LRRC15 and EFNA3 transcripts demonstrated increased expressions across all four molecular subtypes compared to the control group (Figure 4A,B). These results suggest that at least two transcripts, LRRC15 and EFNA3, can be considered as biomarkers for the TNBC group. Consequently, we observed that the four targets had a significant increase in expression in all ages, menopause status, nodal metastasis status, and stages of breast cancer compared to the control (Figure 4E–H, except for CA12 at stage IV, where the increase in expression was not significant. For ethnicity, TSPAN13 and CA12 are overexpressed in Caucasians and Asians in comparison with African Americans. According to tumor histology, LRRC15 is highly expressed in mixed tumors compared to infiltrating ductal, infiltrating lobular, mucinous, medullary, and metaplastic tumors. TSPAN13 is overexpressed in infiltrating ductal, lobular, mixed, and mucinous tumors. EFNA3 is overexpressed in infiltrating ductal and mixed tumors compared to infiltrating lobular, mucinous, and metaplastic tumors, while CA12 presents a low expression in metaplastic and medullary tumors. Regarding the TP53 mutation status, TSPAN13 and CA12 are overexpressed in TP53-non-mutant patients, and EFNA3 is highly expressed in TP53-mutant patients (Appendix A). Therefore, all four proteins have the potential to be candidates for targeted therapy and can aid in precision diagnosis.

### 3.5. LRRC15, EFNA3, TSPAN13, and CA12 as Potential Novel Biomarkers for BC

We further evaluated the diagnostic value of the four transcripts by an ROC curve analysis, determining the sensitivity, specificity, and accuracy. The ROC analysis shows that the AUC of LRRC15 is 0.911 (95% confidence interval (CI) = 0.9030 to 0.9361, sensibility = 87%, specificity = 80%, and accuracy = 86%), as shown in Figure 5A. The corresponding AUC value of EFNA3 is 0.773 (95% CI = 0.7423 to 0.8132, sensibility = 70%, specificity = 75%, and accuracy = 71%) (Figure 5B). The AUC value of TSPAN13 is 0.826 (95% CI = 0.8124 to 0.8622, sensibility = 80%, specificity = 80%, and accuracy = 69%) (Figure 5C). The AUC of CA12 is 0.715 (95% CI = 0.6985 to 0.7552, sensibility = 70%, specificity = 68%, and accuracy = 81%) (Figure 5D). Moreover, the combination of at least two targets increased the detection of breast tumor samples, with 96% sensitivity and 99% specificity when combining LRRC15 and EFNA3; 96% sensitivity and 96% specificity for the combination of LRRC15 and TSPAN13; and 94% sensitivity and 95% specificity when combining LRRC15 and CA12 (Table 2).

### 3.6. Cross-Cancer Overexpression beyond Breast Tumors

To determine whether the identified transcripts were specifically overexpressed in BC, we conducted a pan-cancer analysis. This approach allowed us to identify common changes across various cancer tissues while also characterizing specific alterations within each cancer type. We assessed the mRNA expression levels of LRRC15, EFNA3, TSPAN13, and CA12 in diverse cancer tissues and compared them to normal tissue using the TIMER2.0 database (Figure 6). High levels of LRRC15 mRNA were observed in multiple tumor samples, including breast invasive carcinoma (BRCA) (all molecular subtypes), colon adenocarcinoma (COAD), cholangiocarcinoma (CHOL), glioblastoma multiforme (GBM), head and neck cancer (HNSC), kidney renal papillary cell carcinoma (KIRP), lung adenocarcinoma (LUAD), lung squamous cell carcinoma (LUSC), pancreatic adenocarcinoma (PRAD), rectum adenocarcinoma (READ), stomach adenocarcinoma (STAD), and uterine corpus endometrial carcinoma (UCEC), when compared to their adjacent normal tissues. Similarly, EFNA3 mRNA levels were consistently elevated in most cancers, such as BRCA, bladder urothelial carcinoma (BLCA), cervical squamous cell carcinoma and endocervical adenocarcinoma (CESC), CHOL, COAD, esophageal carcinoma (ESCA), kidney renal clear cell carcinoma (KIRC), KIRP, liver hepatocellular carcinoma (LIHC), LUAD, LUSC, PAAD, STAD, READ, thyroid carcinoma (THCA), and uterine corpus endometrial carcinoma (UCEC), compared to adjacent normal tissues. TSPAN13 mRNA expression levels were also increased in several tumor tissues, including BRCA (except TNBC), BLCA, ESCA, HNSC, KIRC, PRAAD, STAD, THCA, and UCED. Additionally, CA12 expression levels are elevated in various tumor tissues, including BRCA (with a high expression limited to luminal BC), CHOL, GBM, kidney chromophobe (KICH), liver hepatocellular carcinoma (LIHC), LUSC, STAD, and THCA (Figure 6). These results demonstrate that LRRC15, EFNA3, TSPAN13, and CA12 are not only upregulated in breast tumors, but also in other tumor tissues compared to non-tumor samples, indicating that these four genes may play a potentially pivotal role in cancer diagnosis.

We examined the expression levels of the four potential transcripts targets in PBMCs from healthy individuals and their implication for tumor specificity. Our analysis revealed either the absence or low expressions of LRRC15, EFNA3, and CA12 in all cell types, while TSPAN13 exhibited expressions in dendritic cells, platelets, and B cells (Appendix A). Taken together, our results show that the four selected transcripts exhibit low or no expressions in PBMCs in comparison to BC tissues.

### 3.7. Exploring the Signaling Pathways and Protein Interactions of Identified Membrane Proteins in Tumorigenesis

To gain insights into the biological processes and prominent interactions associated with the membrane proteins identified in our study, we examined the potential signaling pathways in which the four selected proteins could be involved. Our findings reveal that most of the proteins identified in the signaling pathways are implicated in various tumorigenesis processes, commonly referred to as cancer hallmarks. In Figure 7A–D and Appendix A, we illustrate several proteins that directly interact with the four selected targets. For instance, LRRC15 interacts with TARDBP, KHDRBS2, NEK2, RARA, ANXA5, DDIT4L, FMOD, GSC, POPDC2, and USP15. According to Gene Ontology annotations, the LRRC15 network proteins play diverse roles in mitotic cell cycle regulation, extracellular matrix organization, signaling, and cell differentiation. EFNA3 interacts predominantly with EPHA family members, along with PRSS23, KRTAP1-1, and LCE2C, and these interactions are mainly functionally associated with cell adhesion, cytoskeleton organization, and cell motility. TSPAN13 interacts with GLP1R and APP, with functional roles in signaling and intracellular protein transport. On the other hand, CA12 interacts with LGALS7, IDH2, PDHX, PDHB, DLAT, PDHA1, IDH3G, IDH3A, and OGDH, all linked to cellular metabolism. The detailed interactions and functions of the identified membrane proteins are presented in Appendix A. Overall, this analysis provides valuable insights into the potential functional roles and significant interactions of the four identified membrane proteins, shedding light on their involvement in tumorigenesis processes.

### 3.8. LRRC15, EFNA3, TSPAN13, and CA12 Are Overexpressed in BC Clinical Samples

To assess and validate LRRC15, EFNA3, TSPAN13, and CA12 in terms of the protein level, immunohistochemistry (IHC) was performed on TMA slides containing 100 cases of invasive breast carcinomas, nine adjacent non-tumor breast tissue samples, and one adjacent breast tumor tissue sample, along with clinicopathological features. The IHC systematic score (0 to +3) was utilized to evaluate protein expression levels by integrating the intensity of positive cells. LRRC15 showed a strong expression in breast tumor tissues (100%), with +2 staining in 1% of the tissues and +3 staining in 99%, while EFNA3 exhibited strong positivity in 90% of the tissues, ranging from 0 (10%), +1 (17%), +2 (22%), to +3 (51%). TSPAN13 demonstrated positive staining in 99% of tissues, with the distribution as follows: 0 (1%), +1 (9%), +2 (9%), and +3 (81%). CA12 displayed a variable staining intensity, with positive staining in 68% of tissues, distributed as follows: 0 (32%), +1 (13%), +2 (14%), and +3 (41%) (Table 3).

BC exhibited significant heterogeneity from the early stages of the disease and was classified into four distinct molecular profiles. Consequently, the expression profiles of LRRC15, EFNA3, TSPAN13, and CA12 proteins were evaluated in different molecular subtypes. The TMA slides comprised 17 samples of the HER2+ subtype, 39 luminal A samples, 27 luminal B samples, and 17 TNBC samples. The strong staining of LRRC15 was observed in all BC tissues, including all molecular subtypes. Intense staining of EFNA3 was seen in 63% of luminal B samples, 59% of TNBC samples, 47% of HER2+ samples, and 41% of luminal A samples. TSPAN13 showed intense staining in 85% of luminal A and luminal B samples, and in 71 and 76% of HER2+ and TNBC samples, respectively. CA12 staining was specific to luminal A and B subtypes, aligning with the results from the transcriptome data. Representative images of tumor samples positive for the four targets are shown in Figure 8. Adjacent non-tumor breast tissue (control) exhibited positive staining for all four targets, albeit at a lower intensity compared to the tumor samples. Regarding BC staging, LRRC15 displayed intense staining in all tissues across the three stages of BC. EFNA3 immunostaining showed strong or moderate staining in 4 out of 6 (67%) tissues at stage I, while 50 out of 72 (70%) and 15 out of 22 (68%) samples exhibited high to moderate staining at stages II and III, respectively. TSPAN13 exhibited high to moderate staining in stage I tissues, 63 out of 72 (87%) samples at stage II, and 21 out of 22 (96%) samples at stage III. CA12 displayed strong to moderate staining in 100% of stage I tissues, with 34 out of 72 (47%) samples showing strong to moderate staining and 38 out of 72 (53%) samples exhibiting weak to no staining at stage II. At stage III, 15 out of 22 (68%) samples displayed intense to moderate staining, while 7 out of 22 (32%) samples exhibited weak staining or an absence of it (Table 3). These results indicate that the target proteins are detected at all stages of the disease, with a predominantly strong staining intensity in early stages.

### 3.9. Validation of LRRC15, TSPAN13, and CA12 Protein Expression Profiles in Breast Cancer Patients: Insights from the CPTAC Dataset

To corroborate our findings, we examined the protein expression patterns of the identified four targets in BC patients using publicly available data from the Clinical Proteomic Tumor Analysis Consortium (CPTAC) dataset from UALCAN. Initially, we analyzed the expression profiles of LRRC15, TSPAN13, and CA12, noting that EFNA3 data were unavailable. Our analysis revealed that LRRC15 and CA12 exhibited increased expressions in BC samples compared to non-tumor samples. However, there was no statistically significant difference in TSPAN13 expression between the tumor and normal samples (Appendix A). The proteomic expression profile data from CPTAC supported our findings regarding the differential protein expressions of LRRC15 and CA12 in BC.

Regarding BC molecular subtypes, LRRC15 showed a high expression in all molecular subtypes compared to non-tumor samples. TSPAN13 exhibited no expression differences in the luminal and HER2+ subtypes, but demonstrated a reduced expression in the triple-negative subtype. CA12 displayed an increased expression exclusively in luminal patients (Appendix A). These findings are consistent with the mRNA expression results (Figure 4D) and immunohistochemistry (Figure 8).

Additionally, we evaluated the expression of these proteins in relation to BC stages: stage I (*n* = 4), stage II (*n* = 74), Stage III (*n* = 32), and normal samples (*n* = 18). LRRC15 and CA12 proteins showed increased expressions in early stage (stage I) BC compared to normal samples. These results suggest that these two selected targets have the potential to be used as early stage biomarkers while still being detectable in advanced stages (II and III) (Appendix A). Furthermore, supporting the findings presented in Table 3, we observed that the targets identified here had high expressions in the early stages of BC. No significant differences in TSPAN13 expression were observed among different tumor stages.

## 4. Discussion

In the past three decades, significant progress has been made in identifying molecular targets associated with cancer, leading to the development of new drugs, and improving diagnostic [25]. However, the current therapeutic approaches for BC patients still have debilitating side effects [6]. In this context, there is an urgent need to develop novel approaches for BC diagnosis and the discovery of targeted therapy strategies based on specific markers. These approaches can contribute to personalized treatment decisions, improving the efficacy of drugs that act on these specific targets and enhancing the overall survival, disease-free survival, and the life quality for patients.

The objective of the current study was to identify differentially expressed genes encoding membrane proteins in tumor versus non-tumor breast tissues. This strategy aims to provide molecular biomarkers to improve molecular diagnosis and accelerate drug design for breast cancer treatment. Membrane proteins play a significant role in cell signaling and extracellular interactions, contributing to tumor formation and progression [16]. These proteins that are overexpressed in tumors are particularly attractive in cancer research due to their broad applicability and accessibility as therapeutic targets [15]. In addition to their direct application in therapy, membrane proteins can be the target to rescue circulating tumoral cells (CTCs), help minimally invasive diagnoses, and can be pursued as a target for the delivery of therapeutic agents, such as nanoparticles and antibody-drug conjugates (ADCs) [26,27,28]. Our data align with the concept of “theragnosis”, which involves combining specific targeted therapy based on accurate diagnostic targets, and emerge as a potential approach for breast cancer treatment.

To identify target membrane proteins for cancer therapy and clinical diagnosis improvement, our study utilized TCGA database, which offered a wide range of omics and clinical data. We focused on paired RNA-seq data, comparing expressions in breast tumor tissues with adjacent non-tumor breast tissues, as well as other healthy tissues. Additionally, we compared the expression profiles with those of PBMCs to identify specific targets relevant to breast tumors. As a result, we identified four membrane proteins—LRRC15, EFNA3, TSPAN13, and CA12—that demonstrated high expressions in BC tissues compared to adjacent non-tumor breast tissues and other healthy samples, with a few exceptions.

Leucine Rich Repeat Containing 15 (LRRC15) is a membrane protein characterized by an extracellular domain, a transmembrane domain, and a short cytoplasmic region. Physiologically, LRRC15 enables collagen, fibronectin, and laminin binding activity, and it is involved in the negative regulation of protein localization to the plasma membrane [29,30,31]. Previous studies have identified LRRC15 as being primarily expressed in astrocytes in response to pro-inflammatory cytokines [32]. Subsequently, it was reported that LRRC15 mRNA exhibited a high expression exclusively in breast tumor tissues, with a weak expression observed in other healthy tissues, except for placental tissue [33]. LRRC15 has emerged as a marker for cancer-associated fibroblasts, and numerous studies have demonstrated its high expression in various solid tumor types, including triple-negative breast cancer, head and neck cancer, non-small-cell lung cancer, pancreatic cancer, ovarian cancer, and osteosarcoma [34,35,36,37,38]. ABBV-085 specifically targets LRRC15, and this drug is currently undergoing a first-in-human phase I study (NCT02565758) for the treatment of sarcomas and other advanced solid tumors [39]; the preliminary results demonstrate its effectiveness in terms of antitumor activity.

The protein Ephrin-A3 (EFNA3), identified in this study, belongs to a large family of cell surface ligands that play a crucial role in regulating a number of biological processes by modulating cell adhesion and interacting with a diverse group of Eph receptor tyrosine kinases [40]. Notably, Gómez-Maldonado et al. (2015) [41] demonstrated through in vitro and in vivo models that EFNA3 expression is associated with metastatic dissemination. Although they did not find a correlation between EFNA3 expression and vascularization, as observed with other members of the ephrin family, the authors showed that EFNA3 was induced by HIF, leading to an increased protein expression in tumor cell lines. Furthermore, their in vivo findings revealed that breast tumors positive for EFNA3 exhibited an increased likelihood of metastasis to other organs [41]. Interestingly, we also observed an elevated expression of EFNA3 in metastatic samples. Recently, Liang et al. (2021) published a study demonstrating that the high mRNA expression of EFNA3 was associated with worse recurrence-free survival (RFS) in BC patients [42].

Tetraspanins are a family of membrane proteins that can interact directly or indirectly with a variety of proteins, including integrins and immunoreceptors. They play a critical role in cell signaling and apoptosis [43,44,45,46,47]. TSPAN13, also known as NET6, has been identified as a breast cancer suppressor gene in the literature. Specifically in breast tumors, there are few publications on the expression profile and functional role of TSPAN13, and the published data are contradictory. Huang et al. (2007) conducted in vitro and in vivo experiments demonstrating that the ectopic expression of TSPAN13 in MDA-MB-231 cells reduced growth and invasion. They also discovered its involvement in pro-apoptotic signaling, leading to increased apoptosis in BC cells and decreased expressions of MMP-1 and MMP-3 [48]. However, an increased expression of TSPAN13 in breast tumors compared to benign tissue has been considered a potential new biomarker for BC and a helpful agent in clinical outcomes [49].

CA12 is a widely expressed enzyme that catalyzes the hydration of carbon dioxide to bicarbonate [50]. Previous studies have shown that the expression profile of CA12 is significantly higher in breast tumor samples compared to normal samples and is associated with estrogen receptor positivity [51,52]. Franke et al. (2020) elucidated an indirect regulatory mechanism where ERα-positive cell lines upregulated CA12 expression through a distal estrogen-responsive region [53]. In the present study, we observed a strong association between positive CA12 staining and a positive estrogen receptor status in primary tumors. Huang et al. (2021) investigated [50] the effect of CA12 silencing on paclitaxel-resistant breast cancer cells (MCF7-TaxR) and found that silencing CA12 activated the mitochondrial apoptosis pathway and promoted apoptosis. The authors concluded that targeting CA12 could be an interesting approach for the treatment of paclitaxel-resistant breast cancer [54]. Other studies have looked for cell surface targets with differential expressions, such as Ziegler and collaborators (2018), who published a study where they looked for potential targets in the plasma membrane in different breast tumor cell lines through mass spectrometry, with the objective of identifying changes in protein expressions for each molecular subtype of BC and, consequently, identified potential targets for personalized treatment [55]. The difference between the studies was that we performed the search directly on the tumor tissues of patients, and that, preferably, it included all the patients with BC.

Based on the extensive research on all targets, we conducted a comprehensive analysis, including TCGA, TNMplot, UALCAN, and HPA databases. Furthermore, we conducted immunohistochemistry validations of breast tissues. The results consistently demonstrated a significant upregulation of mRNA and protein expressions of the four targets in BC samples compared to normal breast and other tissues. The high point is that LRRC15, EFNA3, CA12, and TSPAN13 showed a positive correlation with BC metastasis.

Therefore, we analyzed the expression of these membrane proteins in different molecular subtypes to facilitate a more precise diagnostic approach and achieve satisfactory responses tailored to the specific stages of BC patients. These membrane proteins play crucial roles in tumor biology and hold the potential for diverse clinical applications, primarily by providing specificity for diagnosis and subsequently improving the quality of life for BC patients. Afterwards, the BC samples were clustered according to molecular subtypes, revealing that the TSPAN13 transcript exhibited significant differential expressions in the luminal A, luminal B, and HER2+ subtypes compared to the control. Moreover, the CA12 transcript was found to be overexpressed specifically in luminal A and luminal B patients’ samples. Interestingly, both LRRC15 and EFNA3 showed significant overexpressions in all subtypes, including TNBC, when compared to non-tumor breast tissue. Notably, although we did not observe TSPAN13 mRNA expression in triple-negative samples, we observed a substantial increase in protein expression in TNBC cell lines and particularly in tissue samples. Therefore, TSPAN13 is presently considered as a diagnostic and therapeutic target for triple-negative breast cancer. TNBC remains a significant clinical challenge due to the absence of specific targets, leading to conventional chemotherapy treatment and poorer outcomes [36]. Our findings suggest that LRRC15, EFNA3, and TSPAN13 can be used as potential therapeutic targets for novel drug development, addressing the lack of specificity and high toxicity rates associated with the current treatment options.

To evaluate the performance of the four targets in a diagnostic test, we conducted an ROC curve analysis and determined the sensitivity, specificity, AUC, and accuracy. Remarkably, LRRC15 exhibited a good performance, while EFNA3, TSPAN13, and CA12 showed reliable performances to diagnosis BC. To further improve the test’s specificity, sensitivity, and accuracy, we proposed a combination of at least two proteins, thereby increasing the positive detection rate among patients expressing one or both proteins. The combination of LRRC15 and EFNA3 demonstrated 96% sensitivity and 99% specificity, LRRC15 and TSPAN13 showed 96% sensitivity and 96% specificity, and LRRC15 and CA12 exhibited 94% sensitivity and 95% specificity. The four genes combination included the diagnosis of 100% BC patients. Our study provided comprehensive evidence supporting the significance of four membrane proteins as potential biomarkers and therapeutic targets in BC across all molecular subtypes and tumor stages. Here, we described a multigene panel of clinical utility, which significantly improve the diagnostic accuracy and facilitated personalized treatment approaches.

The study of cell signaling pathways in cancer progression is a crucial step in the development of diagnostic and therapeutic methods for cancer, while also providing valuable insights into the biology of the disease [56,57]. Our analysis revealed the function of LRRC15 enriched in the collagen metabolic process and cell adhesion signaling pathway in breast cancer, showing the importance of LRRC15 for the modulation of the BC microenvironment and tumoral progression. The function of EFNA3 is related to the modulation of cellular metabolism in BC patients, while TSPAN13 enriched in Golgi vesicle transport shows that this protein is related to mediating signal transduction events that play a role in the regulation of cell development, activation, growth and motility. Finally, CA12 enriched in microtubule-based movement-related pathways indicates the functional role of this protein in BC tumoral progression, specially related to cell migration (Appendix A). In this regard, our analysis indicates that LRRC15, EFNA3, TSPAN13, and CA12 are molecular targets and related mechanisms that should be addressed to inhibit BC tumoral progression.

In the subsequent validation phase, we examined the protein expression of selected membrane proteins in a TMA composed of 100 breast tumor samples and nine adjacent non-tumor tissues. Consistent with these previous findings, the TMA analysis revealed the intense staining of LRRC15, EFNA3, and TSPAN13 in BC tissues across all molecular subtypes. On the other hand, CA12 expression was observed only in tissue samples from luminal A and luminal B subtypes, which aligned with previous research indicating that CA12 was significantly highly expressed in tumor samples and correlated with estrogen receptor expression [52]. Consequently, we propose that CA12 is an interesting candidate for targeted therapy in luminal A and luminal B patients, while LRRC15, EFNA3, and TSPAN13 represent potential targets for the treatment of any molecular subtype of BC, particularly TNBC patients. Importantly, the overexpression of LRRC15, EFNA3, TSPAN13, and CA12 was observed in tissues at both early and advanced stages of cancer, indicating that these proteins hold promise as targets for early diagnosis.

## 5. Conclusions

In conclusion, the identified overexpressed membrane proteins provide valuable insights into important molecular events in breast cancer tumorigenesis. They offer potential avenues for targeted therapy tailored to the molecular subtype of BC patients, leading to improved diagnosis and treatment outcomes. Additionally, these membrane proteins could have implications for early diagnosis, making them promising targets for further research and clinical applications in breast cancer management.

## Figures and Tables

**Figure 1 cancers-16-01402-f001:**
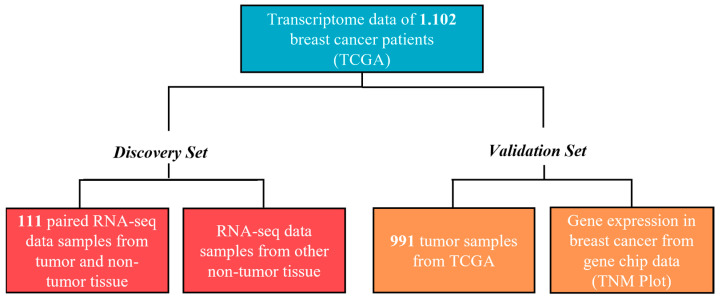
A representative scheme of TCGA cohort analysis groups. Firstly, transcriptome data (RNAseqV2) from 1.102 cases of BC were acquired from TCGA repository. The analysis was divided into a discovery set and a validation set. The discovery set cohort comprised 111 paired RNA-seq data samples from both BC and non-tumor tissues. Additionally, a second analysis was conducted using transcriptome data from other non-tumor tissues. In the validation set, we analyzed 991 tumor samples from TCGA and a second validation using gene expression from gene chip data in TNMplot.

**Figure 2 cancers-16-01402-f002:**
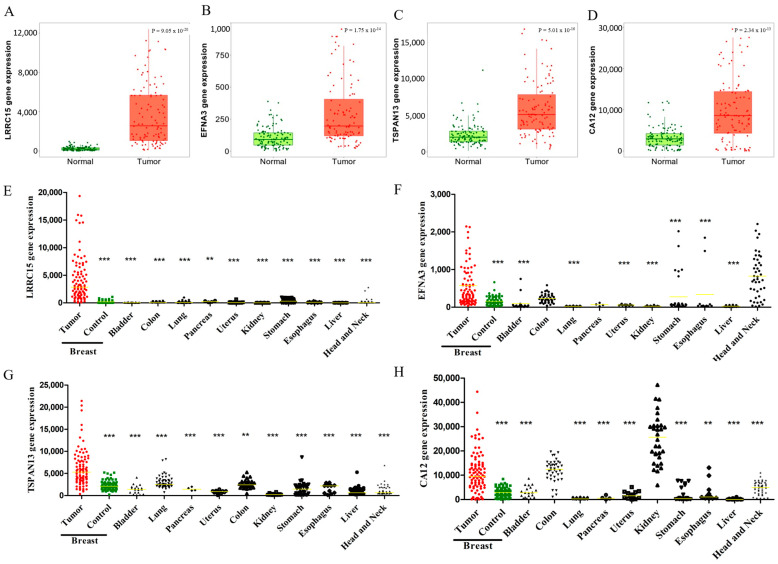
Gene expression profiles of the selected targets across 111 paired tumor samples and their comparisons with non-tumor samples from different tissues. The y-axis represents the gene expression levels (RNAseqV2), and the x-axis represents the samples. Panels (**A**–**D**) show LRRC15, EFNA3, TSPAN13, and CA12 gene expressions, respectively. Green dots represent non-tumor samples from the breast and red dots represent tumor samples. Panels (**E**–**H**) contrast red tumor samples with green or black non-tumor tissues (bladder *n* = 19, lung *n* = 59, pancreas *n* = 4, uterus *n* = 13, colon *n* = 40, kidney *n* = 32, stomach *n* = 32, esophagus *n* = 11, liver *n* = 50, head and neck *n* = 43). The yellow line signifies the median. Statistical analyses involved parametric t-tests for paired samples and non-parametric tests for unpaired samples (*** *p* < 0.001, ** *p* < 0.01).

**Figure 3 cancers-16-01402-f003:**
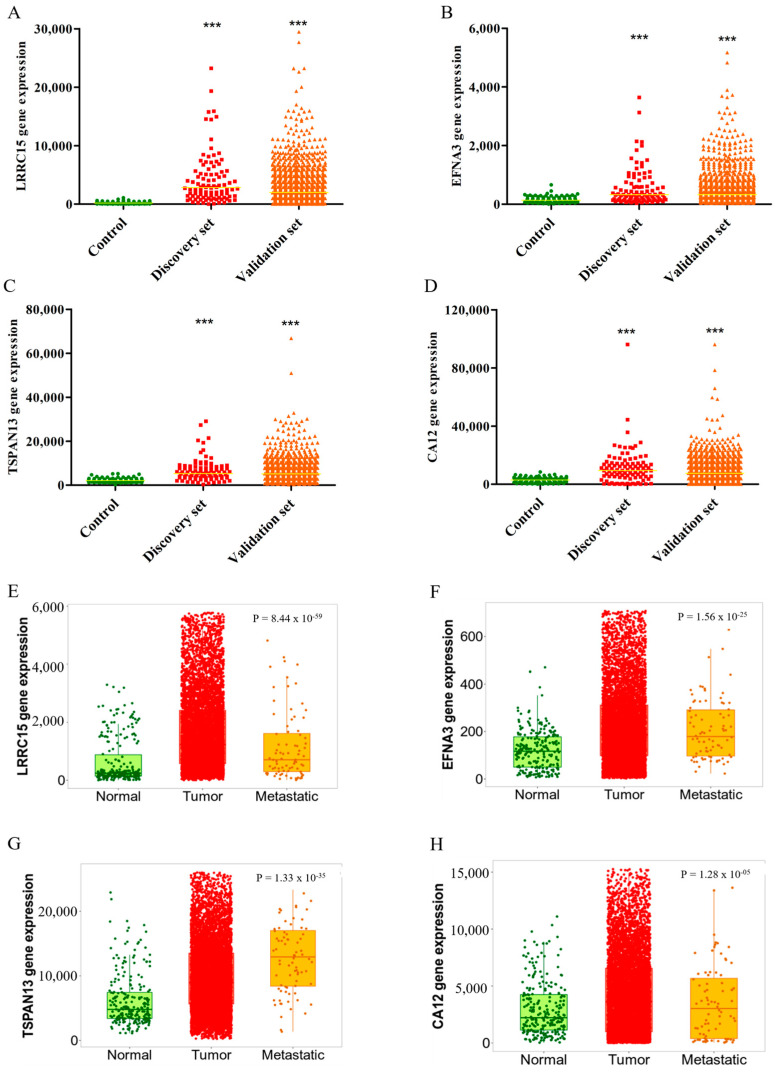
High expression profiles of the four identified targets across discovery and validation cohorts. The y-axis represents the gene expression levels (RNAseqV2). The x-axis represents the samples. Control samples are denoted by green symbols (*n* = 111), paired tumor samples (discovery set) by red symbols (*n* = 111), and total tumor samples (validation set) by orange symbols (*n* = 991). Panels (**A**–**D**) exhibit LRRC15, EFNA3, TSPAN13, and CA12 gene expressions, respectively. Significance is denoted as *** *p* < 0.001, with breast control tissue as the statistical reference. Panels (**E**–**H**) show gene expressions for LRRC15, EFNA3, TSPAN13, and CA12 in breast cancer, respectively, which were analyzed by contrasting normal, tumor, and metastatic samples using gene chip data in TNMplot. Here, the y-axis represents mRNA expression, and the x-axis the study groups. Controls are shown in green, tumor samples in red, and metastatic samples in yellow. Non-tumor breast tissue (control) serves as the statistical reference.

**Figure 4 cancers-16-01402-f004:**
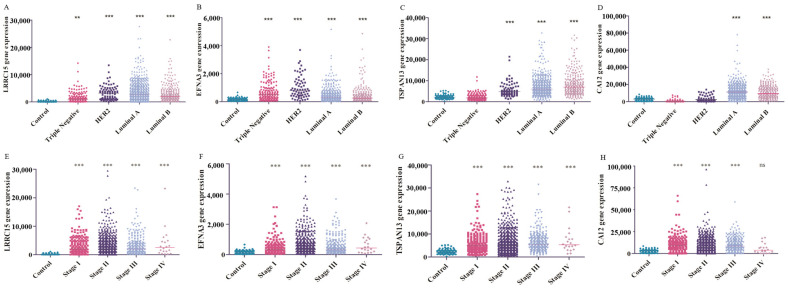
Gene expressions of the four target genes in relation to the molecular subtypes and tumoral stages in BC samples. The y-axis represents the level of expression (RNAseqV2) and the x-axis represents the samples in the different molecular subtypes, stages, and the control sample. Panels (**A**–**D**) exhibit LRRC15, EFNA3, TSPAN13, and CA12 gene expressions, respectively, in different molecular subtypes. Panels (**E**–**H**) shows gene expressions for LRRC15, EFNA3, TSPAN13, and CA12, respectively, in different stages. Significance is denoted as ** *p* < 0.01 and *** *p* < 0.001l mRNA expressions from samples that we used as the breast tissue control (non-tumor) were used as a reference for statistical calculations.

**Figure 5 cancers-16-01402-f005:**
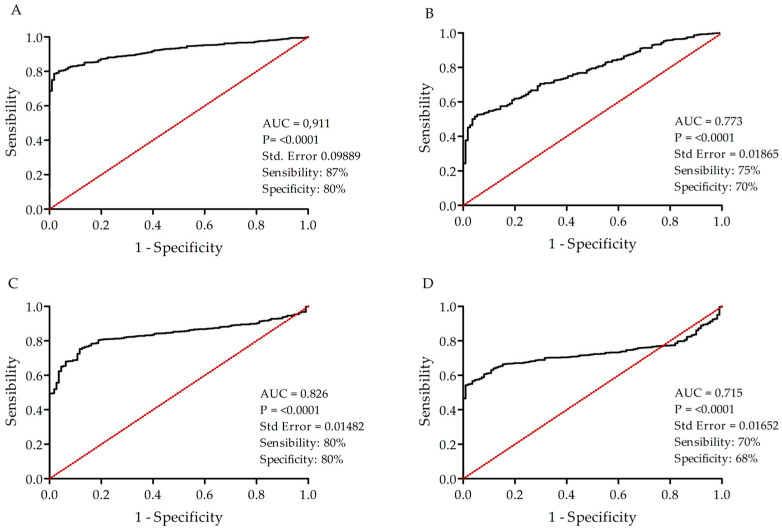
Highly sensibility and specificity values of the LRRC15, EFNA3, TSPAN13, and CA12 transcripts for breast tumors. The ROC curves of the four transcripts for BC were generated using GraphPad Prism 5. Panels (**A**–**D**) exhibit LRRC15, EFNA3, TSPAN13, and CA12 data, respectively. The area under the ROC curve shown in black (AUC), identity line in red, with 95% confidence intervals (CIs) was calculated and differences were considered significant when the *p*-value ≤ 0.05.

**Figure 6 cancers-16-01402-f006:**
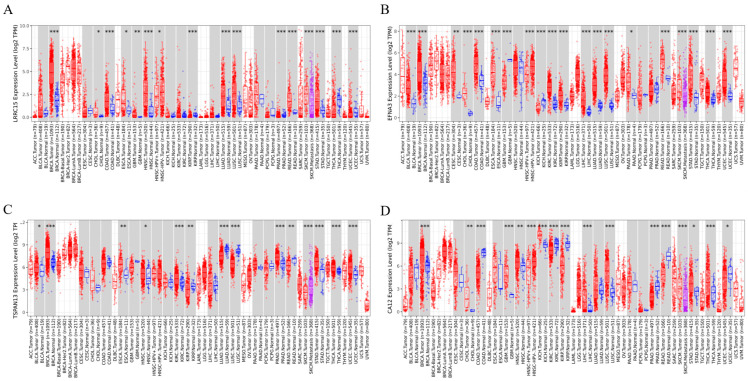
PAN-cancer analysis of the gene expressions of the four transcripts. Differential expressions between tumor and adjacent normal tissue (where available) for (**A**) LRRC15, (**B**) EFNA3, (**C**) TSPAN13, and (**D**) CA12 from TCGA in TIMER 2.0. The red boxplot indicates tumor samples, the blue box plot indicates normal samples, and gray columns indicate comparable paired tumors and adjacent normal tissues. Statistics were performed using the Wilcoxon test; * *p* < 0.05, ** *p* < 0.01, and *** *p* < 0.001 were considered. Adrenocortical Carcinoma (ACC); Bladder Urothelial Carcinoma (BLCA); Breast Invasive Carcinoma (BRCA); Cervical Squamous Cell Carcinoma and Endocervical Adenocarcinoma (CESC); Cholangiocarcinoma (CHOL); Colon Adenocarcinoma (COAD); Diffuse Large B-cell Lymphoma (DLBC); Esophageal Carcinoma (ESCA); Glioblastoma Multiforme (GBM); Head and Neck Cancer (HNSC); Kidney Chromophobe (KICH); Kidney Renal Clear Cell Carcinoma (KIRC); Kidney Renal Papillary Cell Carcinoma (KIRP); Acute Myeloid Leukemia (LAML); Brain Lower-Grade Glioma (LGG); Liver Hepatocellular Carcinoma (LIHC); Lung Adenocarcinoma (LUAD); Lung Squamous Cell Carcinoma (LUSC); Malignant Mesothelioma (MESO); Ovarian Serous Cystadenocarcinoma (OV); Pancreatic Adenocarcinoma (PAAD); Pheochromocytoma and Paraganglioma (PCPG); Prostate Adenocarcinoma (PRAD); Rectum Adenocarcinoma (READ); Skin Cutaneous Melanoma (SKCM); Stomach Adenocarcinoma (STAD); Testicular Germ Cell Tumors (TGCTs); Thyroid Carcinoma (THCA); Thymoma (THYM); Uterine Corpus Endometrial Carcinoma (UCEC); Uterine Carcinosarcoma (UCS); Uveal Melanoma (UVM).

**Figure 7 cancers-16-01402-f007:**
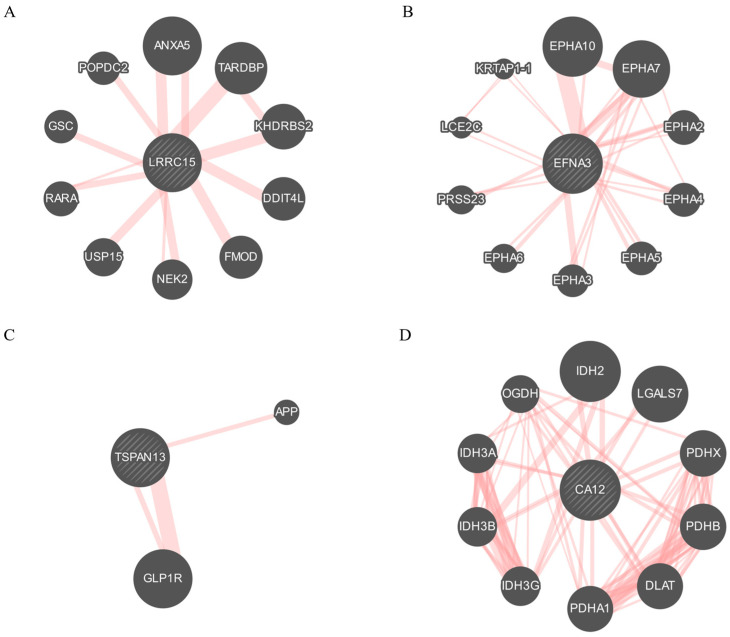
Interactome analysis of LRRC15 (**A**), EFNA3 (**B**), TSPAN13 (**C**), and CA12 (**D**) in breast tumors. Representation of the main proteins involved in the tumorigenesis and tumoral progression processes in the signaling pathways triggered by the four membrane proteins identified in breast tumors. The genes searched are indicated with stripes; the physical interaction is shown by the red line.

**Figure 8 cancers-16-01402-f008:**
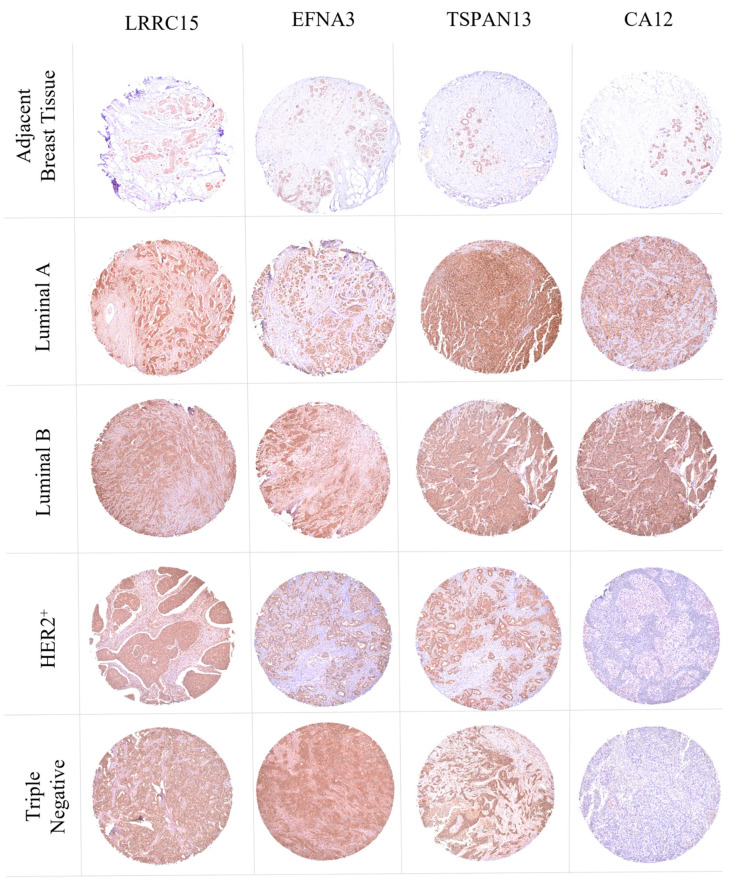
Representative images showing high expressions of LRRC15, EFNA3, TSPAN13, and CA12 proteins across distinct BC molecular subtypes at the protein level. Comprehensive analysis of TMA slides representing distinct molecular subtypes (HER2+, luminal A, luminal B, and TNBC) reveals differential staining intensities for LRRC15, EFNA3, TSPAN13, and CA12. Strong and intense staining patterns are evident across subtypes, with adjacent non-tumor breast tissue serving as a control. Representative images showcase target-specific expressions in breast tumor samples. Photos were taken at 4× magnification.

**Table 1 cancers-16-01402-t001:** Clinical data of BC patients included in the study, including discovery and validation cohorts.

Characteristics	*n*	%
**No. of patients**	1102	100
**Age (years)**	1096	100
**30–50**	332	30
**51–70**	554	50
**71–90**	210	20
**Menopause**	1096	100
**Pre**	230	21
**Post**	742	68
**Unknown**	125	11
**Pathologic T stage**	1102	100
**T1**	284	25
**T2**	640	58
**T3**	138	12
**T4**	40	4
**Pathologic N stage**	1102	100
**N0**	516	47
**N1**	367	33
**N2**	120	11
**N3**	79	7
**NX**	20	2
**Pathologic M stage**	1102	100
**M0**	917	83
**M1**	22	2
**MX**	163	15
**Grade**	1102	100
**I**	186	17
**II**	628	57
**III**	254	23
**IV**	20	2
**Molecular subtype**	1042	100
**Luminal A**	535	51
**Luminal B**	250	24
**HER2**	77	7
**TNBC**	180	17

**Table 2 cancers-16-01402-t002:** Sensitivity, specificity, and accuracy data of combining LRRC15 with the other three selected transcripts.

	Sensibility	Specificity	Accuracy
LRRC15	87%	80%	86%
LRRC15 + EFNA3	96%	99%	96%
LRRC15 + TSPAN13	96%	96%	96%
LRRC15 + CA12	94%	95%	94%

**Table 3 cancers-16-01402-t003:** Analysis of LRRC15, EFNA3, TSPAN13, and CA12 expressions at the protein level in clinical samples.

Staining of BC Specimens					
		LRRC15	EFNA3	TSPAN13	CA12
Positive Samples (%)		100	90	99	68
		Strong	Moderate	Weak	Negative
		**+3**	**+2**	**+1**	**0**
LRRC15		99	1	-	-
EFNA3		51	22	17	10
TSPAN13		81	9	9	1
CA12		41	14	13	32
**Molecular Subtypes (%)**					
		**Luminal A**	**Luminal B**	**HER2+**	**TNBC**
Number of Samples		**39**	**27**	**17**	**17**
LRRC15	+3	100	100	100	96
	+2	-	-	-	4
	+1	-	-	-	-
	0	-	-	-	-
EFNA3	+3	41	63	47	59
	+2	28	15	24	18
	+1	25	19	12	-
	0	5	3	18	24
TSPAN13	+3	85	85	71	76
	+2	8	7	18	6
	+1	8	7	6	18
	0	-	-	6	-
CA12	+3	72	37	19	-
	+2	13	33	-	-
	+1	13	7	6	29
	0	3	22	76	71
**BC Staging (%)**					
		**I**	**II**	**III**	
Number of Samples		**6**	**72**	**22**	
LRRC15	+3	100	99	100	
	+2	-	1	-	
	+1	-	-	-	
	0	-	-	-	
EFNA3	+3	33	50	9	
	+2	33	19	5	
	+1	33	19	27	
	0	-	12	59	
TSPAN13	+3	80	81	86	
	+2	20	7	9	
	+1	-	11	5	
	0	-	1	-	
CA12	**+3**	**80**	**37**	**45**	
	**+2**	20	10	23	
	**+1**	-	16	9	
	0	**-**	**37**	**23**	

## Data Availability

Data are contained within the article and Appendix A.

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
