# Peer review of "Unlocking Overexpressed Membrane Proteins to Guide Breast Cancer Precision Medicine"

_cancers, 2024, doi:10.3390/cancers16071402_

Round 1
Reviewer 1 Report
Comments and Suggestions for Authors
The present manuscript show a strategy for identifying membrane proteins in tumors that can be targeted for specific breast cancer therapy and diagnosis.The proposal of the manuscript is very interesting, showing an integration of gene expression data - bioinformatics analyses, tissue array assays and interactome analyses.
The following points must be considered in the revised version.
- It would be interesting to indicate the different subtypes of breast cancer that represent the 111 tumor samples initially used to identify the differentially expressed genes that encode membrane proteins. How representative is each subtype within these 111 samples?
- In figure 3E-H the authors suggest that the significant increase in the expression of the four target genes in metastatic breast cancer samples could be indicative of biomarkers for clinical progression or as therapeutic candidates for advanced stages of the disease. For this statement, it would be necessary to evaluate the expression of these target genes between non-mestastatic tumor samples versus metastatic samples, in order to show whether expression is differential in disease progression. Is there a statistical difference between these groups - metastatic samples compared to non-metastatic samples?
- The authors indicate that the four targets have a significant increase in expression compared to control samples (Figure 4EH) considering different aspects, including the different stages of disease progression. However, for the CA12 gene this does not seem to occur - there is no statistical difference between the stage IV group and the control. The authors should review this statement.
Furthermore, the increase in CA12 expression appears specific to the Luminal subtype (encompassing subtypes A and B) (Figure 4D), however, there is no statistical difference in the most aggressive degree of the disease (Figure 4H). The authors could explore this aspect in the discussion of the manuscript.
- What are the expression levels of the four target genes in PBMC derived from breast cancer patients? This analysis would be important to compare with expression in healthy individuals (Supplementary Data 2).
- In the literature, it has been described that the TNBC subtype has a high frequency of p53 protein deficiency (mutation and/or loss of expression), related to aggressiveness, drug resistance and poor prognosis of this type of tumor. Interestingly, the authors show that p53WT patients have high expression of TSPAN13 and CA12 - both genes whose expression was not detected in TNBC tumors. The authors should explore this point in the discussion part of the manuscript - expression and activity of p53 x target genes x diagnostic/prognostic markers
However, the Tissue Array data show that 76% of TNBC samples had high protein expression of TSPAN13 (not correlating with TSPAN13 transcript levels). It will be very interesting, and the authors could consider evaluating the expression of p53 in these samples - correlating it with the % of p53WT x p53mut and the activity of this protein.
- Supplemental Table 2 is not included in the files.
Minor points:
In item 2.6, Tissue Microarray Analysis, the categorization of groups based on the percentage of tumor cells exhibiting positive staining needs to be reviewed - group 0 and group 1+ have the same description.
Author Response
Dear Reviewer,
We are very pleased with your report concerning our manuscript “Unlocking Overexpressed Membrane Proteins to Guide Breast Cancer Precision Medicine”. Please find below a point-by-point response to the reviewer’s comments.
The present manuscript show a strategy for identifying membrane proteins in tumors that can be targeted for specific breast cancer therapy and diagnosis. The proposal of the manuscript is very interesting, showing an integration of gene expression data - bioinformatics analyses, tissue array assays and interactome analyses.
The following points must be considered in the revised version.
- It would be interesting to indicate the different subtypes of breast cancer that represent the 111 tumor samples initially used to identify the differentially expressed genes that encode membrane proteins. How representative is each subtype within these 111 samples?
Answer: Thank you for your valuable feedback and suggestions for the revised version of the manuscript. We have carefully considered your points and have made the necessary updates to address the concerns raised.
The paragraph on page 5, line 208 has been added to provide detailed information about the distribution of breast cancer subtypes within the TCGA database cohort used in our study. Specifically, the discovery set (DS) consisted of 111 paired samples, comprising tumor and non-tumor samples from the same patient. Within these samples, the distribution of breast cancer subtypes was as follows: 55% luminal A, 20% luminal B, 13% HER2-positive, and 12% triple-negative breast cancer (TNBC).
- In figure 3E-H the authors suggest that the significant increase in the expression of the four target genes in metastatic breast cancer samples could be indicative of biomarkers for clinical progression or as therapeutic candidates for advanced stages of the disease. For this statement, it would be necessary to evaluate the expression of these target genes between non-mestastatic tumor samples versus metastatic samples, in order to show whether expression is differential in disease progression. Is there a statistical difference between these groups - metastatic samples compared to non-metastatic samples?
Answer: We have conducted further analysis to evaluate the statistical differences between metastatic and non-metastatic tumor samples.
We have added a paragraph on page 9, line 277, which outlines our validation of the differential expression of the four target genes using the TNMplot database. Our analysis, which compared gene expression across malignant breast tumor, metastatic samples, and non-tumoral samples, consistently confirmed increased expression of LRRC15, EFNA3, TSPAN13, and CA12 in tumor samples. Moreover, metastatic samples exhibited elevated transcript levels compared to non-tumoral samples except for CA12. Additionally, metastatic samples exhibited elevated transcript levels compared to tumoral samples, with significant results observed for LRRC15, TSPAN13, and CA12 as determined by Dunn's test (p < 0.05).
- The authors indicate that the four targets have a significant increase in expression compared to control samples (Figure 4EH) considering different aspects, including the different stages of disease progression. However, for the CA12 gene this does not seem to occur - there is no statistical difference between the stage IV group and the control. The authors should review this statement.
Answer: Thank you for your observation regarding the expression levels of the CA12 gene in stage IV breast cancer patients. We have carefully revised the statement to accurately reflect the data. The paragraph on page 9, line 298 has been updated to highlight that while the four target genes exhibit a significant increase in expression compared to controls, however CA12 expression at stage IV did not show statistical significance.
- Furthermore, the increase in CA12 expression appears specific to the Luminal subtype (encompassing subtypes A and B) (Figure 4D), however, there is no statistical difference in the most aggressive degree of the disease (Figure 4H). The authors could explore this aspect in the discussion of the manuscript.
Answer: We have addressed your suggestion by incorporating additional information into the discussion section of our manuscript.
Specifically, we have added a paragraph on page 18, line 567, highlighting the regulatory mechanisms and clinical implications of CA12 expression in breast cancer. The paragraph emphasizes findings from Franke et al. (2020), which elucidated an indirect regulatory mechanism whereby ERα-positive cell lines upregulate CA12 expression through a distal estrogen-responsive region. Additionally, we underscore the observed strong association between positive CA12 staining and a positive estrogen receptor status in primary tumors.
- What are the expression levels of the four target genes in PBMC derived from breast cancer patients? This analysis would be important to compare with expression in healthy individuals (Supplementary Data 2).
Answer: Thank you for your suggestion. Unfortunately, we don’t have PBMC data from breast cancer patients at this time. However, we agree that investigating gene expression profiles in PBMCs of breast cancer patients would indeed be intriguing and we recognize the significance of such comparisons and will consider them for future investigations.
- In the literature, it has been described that the TNBC subtype has a high frequency of p53 protein deficiency (mutation and/or loss of expression), related to aggressiveness, drug resistance and poor prognosis of this type of tumor. Interestingly, the authors show that p53WT patients have high expression of TSPAN13 and CA12 - both genes whose expression was not detected in TNBC tumors. The authors should explore this point in the discussion part of the manuscript - expression and activity of p53 x target genes x diagnostic/prognostic markers
However, the Tissue Array data show that 76% of TNBC samples had high protein expression of TSPAN13 (not correlating with TSPAN13 transcript levels). It will be very interesting, and the authors could consider evaluating the expression of p53 in these samples - correlating it with the % of p53WT x p53mut and the activity of this protein.
Answer:
We thank the reviewer for the excellent point highlighted. The main objective of the work was to identify genes that encode membrane proteins so that we could apply them in clinical practice, whether in diagnosis, prognosis and/or as a target for therapy. Thus, our filter for bioinformatics analyzes (presented in Figure 1) took into account high levels of expression and the largest number of patients. This greater number of patients means including all patients with breast cancer, even considering all tumor subtypes, staging, grade, or presence/absence of metastases; even knowing that each of these classifications’ present molecular characteristics unique. For this reason, when we correlate the expression data of each of the 4 targets identified in the work, it is not correlated with the clinicopathological characteristics of the patients. Therefore, when we also correlated the expression of p53 (at transcript level) and the expression of each of the transcripts, we did not observe a positive correlation.
We agree with your point of view, about exploring the positive correlation between p53 mutation x p53 activity x protein expression of each of the targets, especially TSPAN13. We will continue our analyzes to explore this potential correlation, focusing on TNBC patients. We would like to publish the data as is to disseminate these 4 targets to the scientific community to potentially accelerate the development of new therapeutic approaches.
- Supplemental Table 2 is not included in the files.
Answer: Thank you for bringing to our attention the absence of Supplemental Table 2 in the submitted files. We apologize for the oversight. We will promptly include Supplemental Table 2 in the supplementary files.
Minor points:
In item 2.6, Tissue Microarray Analysis, the categorization of groups based on the percentage of tumor cells exhibiting positive staining needs to be reviewed - group 0 and group 1+ have the same description.
Answer: Thank you for bringing this matter to our attention. We have added a correct description on page 5, line 193.

Reviewer 2 Report
Comments and Suggestions for Authors
The introduction needs to be enriched.
The role of mRNA expression profiles in breast cancer is extremely important in the fields of oncology and molecular biology. In this context, mRNA expression profiles provide valuable information in several aspects:
Diagnosis and Classification: mRNA expression profiles can help in identifying specific types of breast cancer. For example, differentiating between triple-negative, HER2 positive, or estrogen receptor-positive/negative breast cancer, each of which has different therapeutic approaches.
Prognosis:
ETC ETC
Author Response
Dear Reviewer,
We are very pleased with your report concerning our manuscript “Unlocking Overexpressed Membrane Proteins to Guide Breast Cancer Precision Medicine”. Please find below a point-by-point response to the reviewer’s comments.
Revisor:
The introduction needs to be enriched.
The role of mRNA expression profiles in breast cancer is extremely important in the fields of oncology and molecular biology. In this context, mRNA expression profiles provide valuable information in several aspects:
Diagnosis and Classification: mRNA expression profiles can help in identifying specific types of breast cancer. For example, differentiating between triple-negative, HER2 positive, or estrogen receptor-positive/negative breast cancer, each of which has different therapeutic approaches.
Prognosis:
ETC ETC
Authors: Thank you for your valuable feedback and keen observation regarding the need to enrich the introduction of our manuscript. We sincerely appreciate your efforts in reviewing our work and providing constructive suggestions to enhance its clarity and comprehensiveness.
We have carefully reviewed your recommendation and have incorporated the suggested paragraph into the introduction of our manuscript by incorporating an additional paragraph on page 2, line 76. The added paragraph emphasizes the pivotal role of combining gene expression profiling with established genomic platforms in improving the accuracy of classifying breast cancer subtypes. Specifically, we highlighted the importance of this approach in simplifying the identification of specific patient cohorts that may positively respond to tailored therapeutic interventions, as demonstrated by Neagu et al. (2023).
page 2, line 76 “Additionally, the combination of gene expression profiling with already established genomic platforms can improve the accuracy in classifying BC subtypes and simplify the identification of specific groups that would respond positively to therapeutic approaches adapted to their individual characteristics, thus optimizing patient care and outcomes (Neagu et al 2023).”
Reviewer 3 Report
Comments and Suggestions for Authors
The present manuscript describes the identification of 4 membrane proteins overexpressed in Breast Cancer (BC) that could serve as potential biomarkers for BC diagnosis and therapy.
The authors employed a comprehensive bioinformatics analysis using available data from The Cancer Genome Atlas (TCGA), UALCAN, TNM Plot and LinkedOmics and identified four membrane proteins (LRRC15, EFNA3, TSPAN13, and CA12) highly expressed in BC that play important roles in various cancer-related pathways. In agreement, immunohistochemical tissue microarray analysis in BC clinical samples revealed increased expression of these proteins in tumor tissues across all molecular subtypes compared to adjacent breast tissue.
The methodological approach is adequately described. The authors present their observations in a clear and concise way.
Comment(s):
Page 9, lines 35-38: ‘Supplementary data 1’
Page 12, line 22: ‘Supplementary data 2’
Page 17, lines 1-2, 8 and 15: ‘Supplementary data 2/4/5’
It is not clear what ‘Supplementary data’ refer to? The ‘cancers-2841087-supplementary’ file consists of 1 Table (Supplementary Table 1) and 4 Figures (Supplementary Figures 1-4). The authors should check and clarify the issue.
Author Response
Dear Reviewer,
We are very pleased with your report concerning our manuscript “Unlocking Overexpressed Membrane Proteins to Guide Breast Cancer Precision Medicine”. Please find below a point-by-point response to the reviewer’s comments.
Revisor:
The present manuscript describes the identification of 4 membrane proteins overexpressed in Breast Cancer (BC) that could serve as potential biomarkers for BC diagnosis and therapy.
The authors employed a comprehensive bioinformatics analysis using available data from The Cancer Genome Atlas (TCGA), UALCAN, TNM Plot and LinkedOmics and identified four membrane proteins (LRRC15, EFNA3, TSPAN13, and CA12) highly expressed in BC that play important roles in various cancer-related pathways.
In agreement, immunohistochemical tissue microarray analysis in BC clinical samples revealed increased expression of these proteins in tumor tissues across all molecular subtypes compared to adjacent breast tissue.
The methodological approach is adequately described. The authors present their observations in a clear and concise way.
Comment(s):
Page 9, lines 35-38: ‘Supplementary data 1’
Page 12, line 22: ‘Supplementary data 2’
Page 17, lines 1-2, 8 and 15: ‘Supplementary data 2/4/5’
It is not clear what ‘Supplementary data’ refer to? The ‘cancers-2841087-supplementary’ file consists of 1 Table (Supplementary Table 1) and 4 Figures (Supplementary Figures 1-4). The authors should check and clarify the issue.
Authors: We appreciate your attention to detail and will ensure that we clarify the supplementary data in the manuscript for better understanding by the readers. Thank you for bringing this matter to our attention. We re-upload all Figures and Supplementary datas.